# In Vitro Anti-Influenza A Virus H1N1 Effect of Sesquiterpene-Rich Extracts of *Carpesium abrotanoides*

**DOI:** 10.3390/molecules27238313

**Published:** 2022-11-29

**Authors:** Li Li, Shenghui Yang, Dilu Chen, Zhihuang Wu, Meijun Zhang, Fang Yang, Li Qin, Xiaojiang Zhou

**Affiliations:** 1College of Pharmacy, Hunan University of Chinese Medicine, Changsha 410208, China; 2College of Medicine, Hunan University of Chinese Medicine, Changsha 410208, China; 3Changde Institute for Food Inspection, Changde 415000, China; 4Laboratory of Stem Cell Regulation with Chinese Medicine and Its Application, Hunan University of Chinese Medicine, Changsha 410208, China

**Keywords:** the sesquiterpene-rich extracts of *Carpesium abrotanoides*, sesquiterpene, anti-influenza A virus H1N1, TLR4/MyD88/NF-κB signal pathway

## Abstract

Due to a high content of sesquiterpenes, *Carpesium abrotanoides* has been investigated to fully explore its health-promoting properties. Therefore, this work aimed to assess, for the first time, the anti-influenza A virus H1N1 potential of sesquiterpene-targeted fractions of the herb derived from *C. abrotanoides*. Five compounds, including four sesquiterpenes and one aldehyde, were isolated and identified from the sesquiterpene-rich extracts of *C*. *abrotanoides* (SECA), and the contents of three main sesquiterpenes in the SECA were determined. Furthermore, SECA showed a significant protective effect in the MDCK cells infected with influenza A virus (H1N1) in three different conditions: premixed administration, prophylactic administration, and therapeutic administration. SECA can significantly decrease the mRNA expressions of TLR4, MyD88, NF-κB, TNF-α, and IL-6, as well as the protein expressions of TLR4, MyD88, and NF-κB. This result suggests that SECA can resist the influenza A virus H1N1 through the TLR4/MyD88/NF-κB signal pathway.

## 1. Introduction

Influenza is an acute viral disease that easily spreads among people. Its infection causes substantial morbidity and mortality; particularly, the influenza A virus (IAV) H1N1 caused a worldwide pandemic, jeopardizing public health and resulting in numerous deaths [1,2]. After the 2009 pandemic, outbreaks of H1N1 have continued to cause serious illness and increased mortality, particularly in young adults and children [1]. Between 2010 and 2020, seasonal IAV caused an average of 12,000–52,000 deaths per year in the United States [3]. In particular, there are many reports of H1N1 virus mutations, which may lead to the ineffectiveness or reduced effectiveness of antiviral drugs and vaccines [4]. At present, oseltamivir (Tamiflu) is a major therapeutic drug used for treating the H1N1 virus, but there are neuropsychiatric and other adverse reactions, and there are some reports of drug resistance to oseltamivir [5]. In recent years, Chinese herbs, essential components of traditional medicine in countries around the world, have been widely used to treat the H1N1 virus in China because of certain advantages, such as their low toxicity and few side effects [6]. Therefore, the search for new anti-influenza A drugs in Chinese herbs is a popular research topic.

The pattern-recognition receptor (PRR) of host cells recognizes major pathogen-associated molecular patterns (PAMP), causing an innate immune response [7]. Toll-like receptors (TLRs) are involved in non-specific immunity and play an important role in the signal pathway. TLR4 is a member of the TLR family, related to the infection of various viruses and involved in the occurrence of viral diseases. TLR4 is an important activation pathway in the inflammatory response to viral infection [8]. TLR4 is reported to markedly cluster at the site of IAV–cell interaction on the cytomembrane and determines IAV entry and tissue tropism through MyD88 expression [9]. Moreover, it has been reported that IAV-induced releases of IL-6, IL-8, IL-10, TNF-α, and MMP-9 are MAPK- and NF-κB-dependent [10]. The release of many inflammatory factors exacerbates lung tissue injury, so the inhibition of IAV-induced inflammatory factor release is a target for developing anti-IAV drugs. In hepatitis C virus (HCV) infection, the immune system enhances transcription, improves the expression of TLR4, activates B-cells, increases the secretion of IFN-β and IL-6, and induces inflammatory responses and antiviral effects [11]. Relevant studies have confirmed that the influenza virus in Madin–Darby Canine Kidney (MDCK) passage cell lines can replicate well, and the sensitivity to the influenza virus is high; a large number of studies have selected MDCK cells for influenza experiments, confirming that MDCK is the most suitable cell line for establishing an in vitro influenza A virus infection model [12,13,14]. Studies have explored the population heterogeneity of MDCK cells that enables it to derive high IAV-producing cell clones whose superior virus productivity is stable over long-term culturing [15].

*Carpesium abrotanoides* belongs to the genus *Carpesium* of the Compositae family; it is a Chinese herb widely used to treat bruises in Chinese folk medicine [16]. In the early stage, we found a series of active sesquiterpenes against the H1N1 virus from *C. abrotanoides* [16,17]. To further clarify the anti-influenza A virus activity of sesquiterpene-rich extracts of *C*. *abrotanoides* (SECA) and the possible mechanisms behind it, herein, we report the chemical composition, cell cytotoxicity, anti-influenza A virus H1N1 effect, and mRNA and protein expression of TLR4/MyD88/NF-кB in MDCK cells infected with the H1N1 virus, focusing on SECA.

## 2. Results and Discussion

### 2.1. Chemical Composition of SECA

Five compounds, including four sesquiterpenes, 11(13)-dihydrotelekin (**1**) [18], 4(15)-β-epoxyisotelekin (**2**) [19], carabrone (**3**) [20], telekin (**4**) [21] and one aldehyde, 2-(3′,4′-dihydroxyphenyl)-1,3-benzodioxole-5-aldehyde (**5**) [22], were isolated and identified (Figure 1) from SECA, by comparison of their spectroscopic data with those in the literature. Furthermore, the content of sesquiterpenes in SECA was determined by high-performance liquid chromatography (HPLC), and the results show that the contents of 11(13)-dihydrotelekin, telekin and carabrone were 14.42%, 17.39% and 60.21%, respectively (Figure 2).

### 2.2. Cell Cytotoxicity of SECA on MDCK Cells

SECA showed a certain inhibition effect on the proliferation of MDCK cells. With the increase in drug dose, the measured OD value decreased and the inhibition rate on MDCK cell proliferation increased. The median inhibition concentration (IC_50_) of SECA on MDCK cells was calculated with the SPSS software, the IC_50_ was 20.234 μg/mL, and the maximum non-toxic concentration (TC_0_) was 1.131 μg/mL, which applied to subsequent assays. The results are shown in Table 1.

### 2.3. Determination of Median Tissue Culture Infective Dose (TCID_50_) of Influenza A Virus (H1N1)

Based on the method described by Reed and Muench, the TCID_50_ of influenza A virus H1N1 on MDCK cells was 1:389.04, and 100TCID_50_ = 1:3.89; namely the H1N1 virus was diluted to the ratio of 1:3.89 in the subsequent experiments. The infectivity of influenza A virus on MDCK cells is shown in Table 2.

### 2.4. In Vitro Anti-Influenza A Virus H1N1 Effect of SECA

In the premixed administration condition, compared with the negative control group (NCG), the OD value of the virus-infection model group (VIMG) decreased significantly (*p* < 0.05), suggesting that the MDCK cells had been infected successfully by the H1N1 virus. Compared with the VIMG, the OD value of SECA increased significantly (*p* < 0.05), and the SECA antiviral effective rate (ER) was 51.16%, indicating that SECA has certain antivirus activity. Compared with the Oseltamivir control group (OCG), the OD value of SECA was not significantly different (*p* > 0.05) (Figure 3a).

In the condition with prophylactic administration, compared with the NCG, the OD value of the VIMG decreased significantly (*p* < 0.05), suggesting that the MDCK cells had been infected successfully by the H1N1 virus. Compared with the VIMG, the OD value of SECA increased significantly (*p* < 0.05), and the SECA antiviral ER was 83.29%, indicating that SECA has better antivirus activity. Compared with the OCG, the OD value of SECA was not significantly different (*p* > 0.05) (Figure 3b).

In the condition with therapeutic administration, compared with the negative control group (NCG), the OD value of the VIMG decreased significantly (*p* < 0.05), suggesting that the MDCK cells had been infected successfully by the H1N1 virus. Compared with the VIMG, the OD value of SECA increased significantly (*p* < 0.05), and the SECA antiviral ER was 42.43%, indicating SECA has certain antivirus activity. Compared with the OCG, the OD value of SECA was significantly different (*p* < 0.05) (Figure 3c).

### 2.5. mRNA Expressions of TLR4, MyD88, NF-кB, TNF-α and IL-6 in MDCK Cells Infected by Influenza A Virus H1N1

According to Figure 4, compared with the negative control group (NCG), mRNA expressions of TLR4, MyD88, NF-κB, TNF-α, and IL-6 in the virus infection model group (VIMG) were significantly increased, and the difference was statistically significant (*p* < 0.01). Compared with VIMG, the mRNA expressions of TLR4, MyD88, NF-κB, TNF-α, and IL-6 in low-dose group (LDG), medium-dose group (MDG), high-dose group (HDG) and Oseltamivir control group (OCG) were significantly decreased (*p* < 0.01).

### 2.6. Protein Expressions of TLR4, MyD88, and NF-кB in MDCK Cells Infected by Influenza A Virus H1N1

TLR4, MyD88, and NF-κB protein expressions were significantly increased in the virus infection model group (VIMG), compared with the negative control group (NCG) (*p* < 0.01). According to Figure 5, compared with the VIMG, the protein expressions of TLR4, MyD88 and NF-κB in MDCK cells were significantly reduced in the low-dose group (LDG), medium-dose group (MDG), high-dose group (HDG), and Oseltamivir control group (OCG); the difference was statically significant (*p* < 0.01).

## 3. Materials and Methods

### 3.1. Materials and Reagents

The influenza A virus H1N1 strain (A/PR8/34)-adapted mouse lung was kindly presented by the College of Life Sciences, Hunan Normal University (Changsha, China). It was cultured in 9-day-old chicken embryo and the allantoic fluid was harvested at 3 days after culture, filtered using a 0.22 μm filter, then stored at −80 °C. The H1N1 virus titer was determined by a blood coagulation assay. The Madin–Darby canine kidney (MDCK) cell line was purchased from the National Collection of Authenticated Cell Cultures (Shanghai, China). In total, 0.25% EDTA-trypsin, fetal bovine serum, and Trizol were purchased from Gibco (Thermo Fisher, Waltham, MA, USA). DMEM high-glucose culture medium (containing sodium pyruvate) was purchased from Hyclone (Logan, UT, USA). TPCK–trypsin was purchased from Sigma Chemical Co. (St. Louis, MO, USA). RIPA lysate was purchased from Well Biological Science Co. (Changsha, China). Rabbit anti-dog NF-κB, TLR4, MyD88, mouse anti-dog β-actin, HRP labeled goat anti-rabbit IgG, and goat anti-mouse IgG antibodies were purchased from Proteintech Group, Inc (Chicago, IL, USA). Quantitative real-time polymerase chain reaction (QRT-PCR) primers were synthesized by Sangon Biotech Co., Ltd. (Shanghai, China), and the mRNA reverse transcription kit and Ultra-SYBR Mixtures were purchased from CoWin Bioscience Co., Ltd. (Changsha, China).

### 3.2. Preparation of SECA

The dried herbs of *C*. *abrotanoides* (10 kg) were extracted with 70% ethanol (2 × 80 L) to yield an extract (1120 g), which was suspended in water and partitioned by petroleum ether and EtOAc (each 4 × 4 L), respectively. The EtOAc extracts (170 g) were submitted to macroporous adsorption resin HP 20 column elution with water and 70% ethanol (6 L), respectively. Finally, 70% ethanol eluate was collected and further purified by recrystallization to obtain SECA (68.2 g).

### 3.3. Isolation of SECA

The SECA (1.15 g) were divided into two parts (Frs. 1-1–1-2) by an MCI gel CHP 20P column elution with gradient aqueous MeOH. Fr. 1-2 (0.43 g) was submitted to vacuum liquid chromatography (VLC) on a silica gel column eluted with petroleum ether/EtOAc (3:1, 2:1, 1:1) to afford **1** (18.7 mg), **2** (6.6 mg) and **3** (29.2 mg). Fr. 1-1 (0.18 g) was purified by Sephadex LH-20 (MeOH) to yield **4** (17.0 mg) and **5** (7.3 mg).

### 3.4. Structural Elucidation of Compounds

The isolated compounds were identified using NMR spectra, and all the data are in good agreement with those previously reported.

11(13)-dihydrotelekin (**1**): ^1^H NMR (600 MHz, CD_3_OD): *δ*: 4.84 (1H, br s, H-15a), 4.71 (1H, br s, H-15b), 4.58 (1H, m, H-8), 2.96 (1H, m, H-11a), 2.82 (1H, m, H-7a), 2.67 (1H, m, H-3a), 2.09 (1H, m, H-3b), 1.99 (1H, dd, *J* = 15.4, 4.6 Hz, H-9a), 1.89 (1H, td, *J* = 13.1, 4.9 Hz, H-1a), 1.78 (1H, dd, *J* = 15.4, 1.6 Hz, H-9b), 1.64 (1H, dd, *J* = 13.9, 5.9 Hz, H-6a), 1.58 (2H, m, H-2), 1.47 (1H, dd, *J* = 13.9, 12.4 Hz, H-6b), 1.16 (3H, d, *J* = 7.2 Hz, H-13), 1.08 (1H, td, *J* = 13.1, 3.2 Hz, H-1b), 0.91 (3H, s, H-14); ^13^C NMR (150 MHz, CD_3_OD): *δ*: 182.3 (C-12), 153.1 (C-4), 108.2 (C-15), 80.0 (C-8), 74.6 (C-5), 42.2 (C-11), 38.4 (C-7), 37.9 (C-10), 37.1 (C-1), 36.4 (C-2), 32.8 (C-3), 28.2 (C-9), 22.9 (C-6), 22.1 (C-14), 9.6 (C-13).

4(15)-β-epoxyisotelekin (**2**): ^1^H NMR (600 MHz, CD_3_OD): *δ*: 6.04 (1H, d, *J* = 1.1 Hz, H-13a), 5.62 (1H, d, *J* = 1.0 Hz, H-13b), 4.56 (1H, m, H-8), 3.27 (1H, m, H-7), 2.76 (1H, dd, *J* = 4.3, 2.0 Hz, H-15a), 2.64 (1H, d, *J* = 4.3 Hz, H-15b), 1.07 (3H, s, H-14); ^13^C NMR (150 MHz, CD_3_OD): *δ*: 172.7 (C-12), 143.9 (C-11), 120.9 (C-13), 78.9 (C-8), 74.5 (C-5), 62.1 (C-4), 54.1 (C-15), 38.6 (C-7), 38.3 (C-10), 37.2 (C-9), 35.7 (C-1), 31.0 (C-6), 29.7 (C-3), 22.1 (C-14), 20.6 (C-2).

carabrone (**3**): ^1^H NMR (600 MHz, CDCl_3_): *δ*: 6.09 (1H, m, H-13a), 5.45 (1H, m, H-13b), 4.67 (1H, m, H-8), 3.05 (1H, m, H-7), 2.42 (2H, t, *J* = 7.3 Hz, H-3), 2.20 (2H, m, H-6), 2.04 (3H, s, H-15), 1.47 (2H, m, H-9), 0.96 (3H, s, H-14), 0.83 (2H, m, H-2), 0.34 (1H, m, H-1), 0.26 (1H, m, H-5); ^13^C NMR (150 MHz, CDCl_3_): *δ*: 208.6 (C-4), 170.2 (C-12), 138.7 (C-11), 122.3 (C-13), 75.4 (C-8), 43.2 (C-3), 37.3 (C-7), 36.9 (C-9), 33.8 (C-1), 30.4 (C-6), 29.8 (C-15), 23.0 (C-2), 22.5 (C-5), 17.9 (C-14), 16.8 (C-10).

telekin (**4**): ^1^H NMR (600 MHz, CD_3_OD): *δ*: 6.07 (1H, d, *J* = 1.1 Hz, H-13a), 5.67 (1H, d, *J* = 0.9 Hz, H-13b), 4.83 (1H, t, *J* = 1.5 Hz, H-15a), 4.67 (1H, br s, H-15b), 4.59 (1H, td, *J* = 5.2, 1.1 Hz, H-8), 3.35 (1H, m, H-7), 2.66 (1H, m, H-3a), 2.11 (1H, m, H-3b), 2.05 (1H, dd, *J* = 15.5, 5.0 Hz, H-9a), 1.81 (1H, dd, *J* = 15.5, 1.4 Hz, H-9b), 1.60 (2H, m, H-2), 0.93 (3H, s, H-14); ^13^C NMR (150 MHz, CD_3_OD): *δ*:172.8 (C-12), 152.6 (C-4), 144.2 (C-11), 120.7 (C-13), 108.4 (C-15), 79.1 (C-8), 74.5 (C-5), 39.0 (C-7), 37.5 (C-10), 36.8 (C-1), 36.5 (C-2), 34.7 (C-3), 33.0 (C-9), 23.0 (C-6), 22.2 (C-14).

2-(3′,4′-dihydroxyphenyl)-1,3-benzodioxole-5-aldehyde (**5**): ^1^H NMR (600 MHz, CD_3_OD): *δ*: 9.69 (1H, s, H-10), 7.32 (1H, d, *J* = 8.2 Hz, H-6), 7.30(1H, s, H-4), 6.92 (1H, br s, H-7), 6.90 (1H, br s, H-5′), 6.85 (1H, br s, H-2′), 6.74 (1H, br s, H-6′), 5.22 (1H, s, H-2); ^13^C NMR (150 MHz, CD_3_OD): *δ*: 193.0 (C-10), 153.7 (C-9), 147.2 (C-8), 146.7 (C-1′), 146.1 (C-3′), 131.1 (C-4′), 130.9 (C-5), 126.4 (C-6), 119.4 (C-6′), 116.2 (C-7), 115.8 (C-5′), 115.3 (C-2′), 114.8 (C-4), 104.8 (C-2).

The ^1^H NMR and ^13^C NMR spectra for compounds **1**–**5** are available in the Appendix A.

### 3.5. HPLC Assay

For the HPLC analysis, a WondaSil C18 (4.6 × 150 mm, 5 μm) column (shimadzu, Japan) was used. With isocratic elution, the mobile phases consisted of acetonitrile (mobile phase A) and water (mobile phase B), with a ratio 29.5:70.5%, *v*/*v* for 60 min. The flow rate was 1 mL/min, and the column temperature was maintained at 30 °C. The injection volume was 20 µL, and the detection wavelength was set at 211 nm.

### 3.6. Drug Preparation for Antiviral Assay

In total, 72.7 mg SECA crystal was dissolved in 0.727 mL DMSO and filtered by a 0.2 μm filter. The 100 mg/mL solution was obtained, which could be diluted using culture medium during subsequent assays.

The Oseltamivir phosphate capsule was purchased from Roche pharmaceuticals co. LTD (Shanghai, China). The drug was dissolved in distilled water; the concentration was 5 mg/mL and it was stored at −20 °C.

### 3.7. Cytotoxicity Assay of SECA on MDCK Cells

MDCK cells were routinely digested with 0.25% EDTA-trypsin and adjusted to a density of 1 × 10^5^ cells/mL, then seeded into a 96-well plate (100 μL/well) and cultured at 37 °C under 5% CO_2_ for 24 h. The culture supernatants were discarded and the cells were washed three times with PBS. The SECA storage solution was diluted to 25, 12.5, 6.25, 3.125, and 1.5625 µg/mL using the MDCK cell culture medium (MCCM) containing 90% DMEM, 2.5%HEPES, 1%BSA, 10% fetal bovine serum, 100 U/mL penicillin, and 100 μg/mL streptomycin. Each concentration of drug was added to the MDCK cells (150 µL/well), and six duplicate wells were prepared for each concentration. Cells added with only 150 µL of MCCM were set as the negative control group (NCG) and cultured at 37 °C under 5% CO_2_. After 44 h of incubation, the 15 µL of CCK8 reagent was added into each well and the cells were continually cultured for another 4 h. The absorbance of each well at the 450 nm wavelength was measured with a spectrophotometer (Bio-tek, Santa Clara, CA, USA). The cell growth inhibition rate (IR) was calculated for different concentrations of drug according to the formula “IR = 1 − (average OD value of different concentration drug)/(average OD value of NCG)”, and the median inhibition concentration (IC_50_) and maximum nontoxic concentration (TC_0_) of the drug in relation to MDCK cell proliferation were calculated using Probit regression analysis in the statistical software SPSS 21.0.

### 3.8. Determination of Median Tissue Culture Infective Dose (TCID_50_) of Influenza A Virus (H1N1) Using Cytopathic Effect (CPE) Assay

MDCK cells were routinely digested with 0.25% EDTA-trypsin, adjusted to 1 × 10^5^ cells/mL, and seeded into a 96-well plate (100 µL/well) and then cultured at 37 °C under 5% CO_2_ for 20–24 h; afterwards, the supernatants of MDCK cells were discarded and the cells were washed 2 times with PBS. The H1N1 virus was successively diluted to six different concentration of 10^−1^, 10^−2^, 10^−3^, 10^−4^, 10^−5^, and 10^−6^ using virus growth medium (VGM) containing 90% DMEM, 2.5% HEPES, 1% BSA, 2 μg/mL TPCK-trypsin, 100 U/mL penicillin, and 100 μg/mL streptomycin; then the different concentration dilutions of H1N1 virus were added to the MDCK cells (100 µL/well). Those MDCK cells without added virus were set as the negative control group (NCG), and 6 duplicates were set for each group and all the cells were cultured at 37 °C under 5% CO_2_ for 2 h. Subsequently, the H1N1 virus solution was discarded and the cells were washed 2 times with PBS solution; then 150 μL virus growth medium (VGM) was added to each virus dilution group and NCG. After further incubation for 3 to 5 days, the H1N1 virus cytopathic effect (CPE) on MDCK cells was observed daily. The TCID_50_ of the H1N1 virus on MDCK cells was determined by the method described by Reed and Muench.

### 3.9. Determination of Anti-Influenza A Virus H1N1 Effect of SECA In Vitro

In order to explore the anti-influenza A virus effect of SECA, three different conditions were employed, including premixed administration, prophylactic administration, and therapeutic administration, and the MDCK cells were prepared via the method described in Section 3.8.

Under premixed administration, equal volumes of mixtures containing 2 × TC_0_ SECA and 2 × 100TCID_50_ virus solution were prepared and laid at 4 °C for 24 h, then subsequently added to 96-well cell plates (200 μL/well) and cultured at 37 °C, 5% CO_2_ for 2 h. Following the discarding of the supernatants, MDCK cells were washed 2 times with PBS, and the virus growth medium (VGM) was added into the 96-well plate (200 μL/well), for 72 h of incubation. Those cells added with only 100TCID_50_ virus were set as the virus-infection model group (VIMG), those without adding SECA or virus as the negative control group (NCG), and those with 5 μg/mL Oseltamivir phosphate and 2 × 100TCID_50_ as the Oseltamivir control group (OCG). Under prophylactic administration, TC_0_ SECA was added to a 96-well plate (200 μL/well) and cultured at 37 °C with 5% CO_2_ for 2 h. Subsequently, the supernatants were discarded, the cells were washed 2 times with PBS, and 100 TCID_50_ virus solution was added to the plate (100 μL/well) for incubation for 2 h. Afterwards, the virus solution was discarded, and MCCM (200 μL/well) was added onto the plate and submitted for 72 h of incubation at 37 °C with 5% CO_2_. VIMG and NCG were established as above, with only adding 2.5 μg/mL Oseltamivir phosphate to form the Oseltamivir control group (OCG). Under therapeutic administration, the 100TCID_50_ virus solution was added onto a 96-well plate (100 μL/well) and the cells were cultured at 37 °C with 5% CO_2_ for 2 h. After the virus solution was discarded, the cells were washed 2 times with PBS. Then TC_0_ SECA was added to the plate (200 μL/well) and the cells were cultured at 37 °C with 5% CO_2_ for a further 72 h. VIMG, NCG, and OCG were established as above.

After 68 h of incubation, the CCK reagent was added onto the 96-well plate (20 μL/well) and cultured for 4 h. Afterwards, the optical density (OD) value of MDCK cells at the 450 nm wavelength was measured with a spectrophotometer. Cell survival rate (SR) and antiviral effective rate (ER) were determined according to the formulae “SR(%) = (average OD value of different drug group)/(average OD value of NCG) × 100%”, and “ER(%) = (average OD value of different drug group − average OD value of VIMG)/(average OD value of NCG − average OD value of VIMG) × 100%”, respectively.

### 3.10. Effect of SECA on Expressions of mRNA and Protein of TLR4, MyD88, NF-кB, TNF-α, and IL-6 in MDCK Cells Infected by Influenza A Virus H1N1

To observe the effect of SECA on the expression of the mRNA and protein of TLR4, MyD88, NF-кB, TNF-α, and IL-6 in MDCK cells infected by H1N1 virus, the cells were prepared via the method described in Section 3.8. The supernatants were discarded and cells were washed 2 times with PBS. Subsequently, the 100TCID_50_ virus solution was added to the 96-well plate (100 μL/well) and incubated at 37 °C with 5% CO_2_ for 2 h. Then, the virus solution was discarded, and the cells were washed 2 times with PBS. Then the SECA of the TC_0_ (high-dose group, HDG), 1/2 TC_0_ (medium-dose group, MDG), and 1/4 TC_0_ (low-dose group, LDG) concentrations were diluted using VGM and added onto the plates (200 μL/well); the cells were cultured at 37 °C with 5% CO_2_ for a further 48 h. Cells without added virus were set as the negative control group (NCG), adding only H1N1 virus as the virus-infection model group (VIMG) and adding only 2.5 μg/mL Oseltamivir phosphate as the Oseltamivir control group (OCG). After incubation for 48 h, the supernatants were discarded, the cells were gently washed 3 times with ice PBS, and the total RNA and total protein of the cells were extracted.

### 3.11. Quantitative Real-Time Reverse Transcriptase PCR (qRT-PCR)

Total RNA was extracted using Trizol reagent (Invitrogen, Waltham, MA, USA) according to the manufacturer’s instructions. Afterwards, the extracted total RNA was reverse transcribed into cDNA strands using the RevertAid First cDNA Synthesis Kit (Thermo Scientific, Waltham, MA, USA). The cDNA was then amplified using the CoWin Bioscience Co., Ltd. (Changsha, China), and the primer sequences for each gene are shown in Table 3. Briefly, each 30 μLof reaction volume contained 15 μL of SYBR Green PCR Master Mix, 2 μL of each primer (0.4 μM), 11 μL of diethyl pyrocarbonate (DEPC)-treated water and 2 μL template.

The optimum conditions for the PCR amplification of the cDNA were established by following the manufacturer’s instructions. The PCR cycle conditions were 95 °C, pre-denaturation for 10 min, then 95 °C for 15 s and 60 °C for 30 s, repeated for 40 cycles. The data were analyzed using StepOne software (Applied Biosystems), and the cycle numbers at the linear amplification threshold values (Ct) for the endogenous dog β-actin gene and the target genes were recorded. Relative gene expression (target gene expression normalized to the expression of the endogenous β-actin gene) was calculated using the comparative Ct method (2^−∆∆Ct^). The experiments and analysis were conducted three times independently.

### 3.12. Western Blot Analysis

MDCK cells were lysed with RIPA lysate (Well Biological Science Co., Ltd. Changsha, China) containing protease and phosphatase inhibitors (Merck, Germany), and total proteins were extracted according to the manufactory’s instructions. Protein concentration was measured by a BCA protein quantitation kit (Well Biological Science Co., Ltd. Changsha, China). Volumes equivalent to 100 μg of protein were transferred onto 10% polyacrylamide and electrophoresis was carried out. Separated proteins were transferred onto an NC membrane (Millipore, Merck) via the Semi-Dry Transfer Unit. Next, blots were blocked with 5% skim milk and 0.05% Tween-20 for 2 h at room temperature and independently incubated overnight at 4 °C with specific primary antibodies (Table 4). HRP-labeled goat anti-rabbit IgG (1:6000) and goat anti-mouse IgG (1:5000) were diluted with 1 × TBS-tween-20 (TBST), and incubated with the membrane at room temperature for 90 min. Subsequently, the NC membrane was washed 3 times with TBST. Protein development was performed with the western ECL chemiluminescent solution (Pierce, Waltham, MA, USA), and the X film was imaged. β-actin was used as an internal control to normalize protein expression levels. The images were digitized, and the image Pro-Plus software was used to analyze the gray value of the target protein band in the scanned image. The relative ratio of grey value from certain target proteins to the β-actin band was considered the relative expression of the target protein.

### 3.13. Statistical Analysis

All results are expressed as mean values ± standard error. Comparisons are based on an analysis of variance between the two groups by one-way ANOVA. Differences were considered statistically significant when *p* < 0.05.

## 4. Discussion and Conclusions

The influenza A virus is one of the most persistent and unpredictable human pathogens. Influenza continues to cause regular seasonal epidemics, unpredictable pandemics, and frequent and deadly zoonotic outbreaks worldwide. Currently, anti-influenza A virus drugs have certain side effects, and drug-resistant mutants commonly emerge, so there is an urgent need to develop novel anti-influenza A virus drugs. Our preliminary study showed that SECA have good activity against influenza A virus H1N1 in chicken embryonated eggs. The current study shows that SECA not only exert good antiviral activity in three different conditions but can also decrease the mRNA and protein expressions of TLR4, MyD88, and NF-κB. Thus, this result suggests SECA may serve as a potential anti-influenza A H1N1 agent in the future.

TLRs are important parts of the human innate immunity against pathogens. Currently, the research on the association with influenza A virus infection mainly focuses on TLR4 and TLR7 [23]. TLR4 is a typical pattern-recognition receptor of TLRs [24]. TLR4 mainly recognizes PAMPs, such as bacterial lipopolysaccharides and the envelope proteins of viruses. When TLR4 binds to PAMP, its intracellular segment will recruit MyD88, which is a key junction molecule of the TLRs’ signal transduction pathway [25]. Then, the death domain of MyD88 binds to the IL-1-receptor-related kinase and recruits and activates downstream TRAF6 and NF-κB. IL-6, TNF- α, and other inflammatory factors are further activated [26,27,28]. Therefore, the TLR4/MyD88/NF-κB signal pathway plays a very important role in the process of influenza A. Furthermore, the expression of many inflammatory factors induced by the influenza virus also leads to tissue and cell damage [29]. The experimental results show that SECA can significantly downregulate the expressions of TLR4, MyD88, and NF-κB in MDCK cells, and reduce the secretion of inflammatory factors such as IL-6 and TNF-α, thereby alleviating the damages caused by IAV. Therefore, this result suggests that SECA can resist influenza A virus H1N1 through the TLR4/MyD88/NF-κB signal pathway.

In conclusion, the present study first demonstrated in vitro the potential anti-influenza A virus H1N1 effect of sesquiterpene-rich extracts of *C*. *abrotanoides*. Their effects directly involved the contents of the sesquiterpenes. Our finding offers a new herbal medicine as a potential anti-influenza A (H1N1) agent. However, the safe usage and application of these sesquiterpene-rich extracts should be further investigated in animal models.

## Figures and Tables

**Figure 1 molecules-27-08313-f001:**
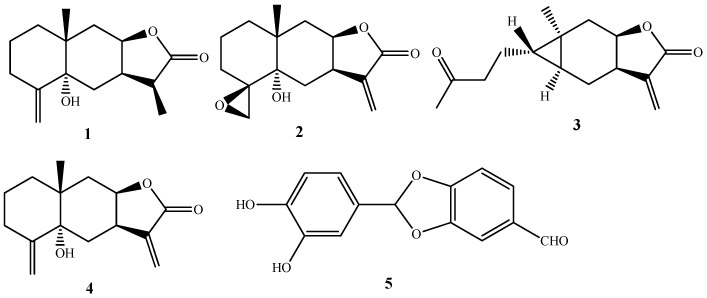
Structures of compounds **1**–**5**. 11(13)-dihydrotelekin (**1**), 4(15)-β-epoxyisotelekin (**2**), carabrone (**3**), telekin (**4**) and one aldehyde, 2-(3′,4′-dihydroxyphenyl)-1,3-benzodioxole-5-aldehyde (**5**).

**Figure 2 molecules-27-08313-f002:**
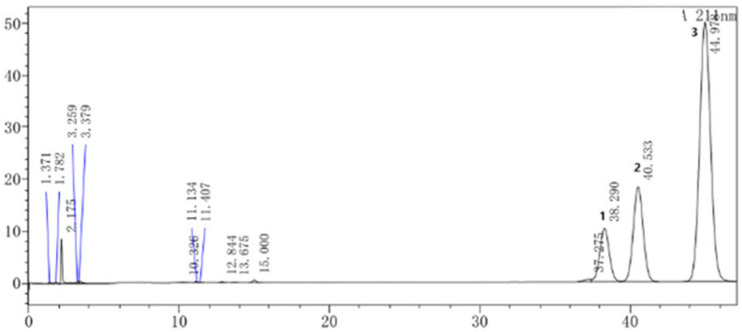
HPLC chromatogram sesquiterpenes with different compounds: 11(13)-dihydrotelekint (**1**), telekin (**2**), carabrone (**3**).

**Figure 3 molecules-27-08313-f003:**
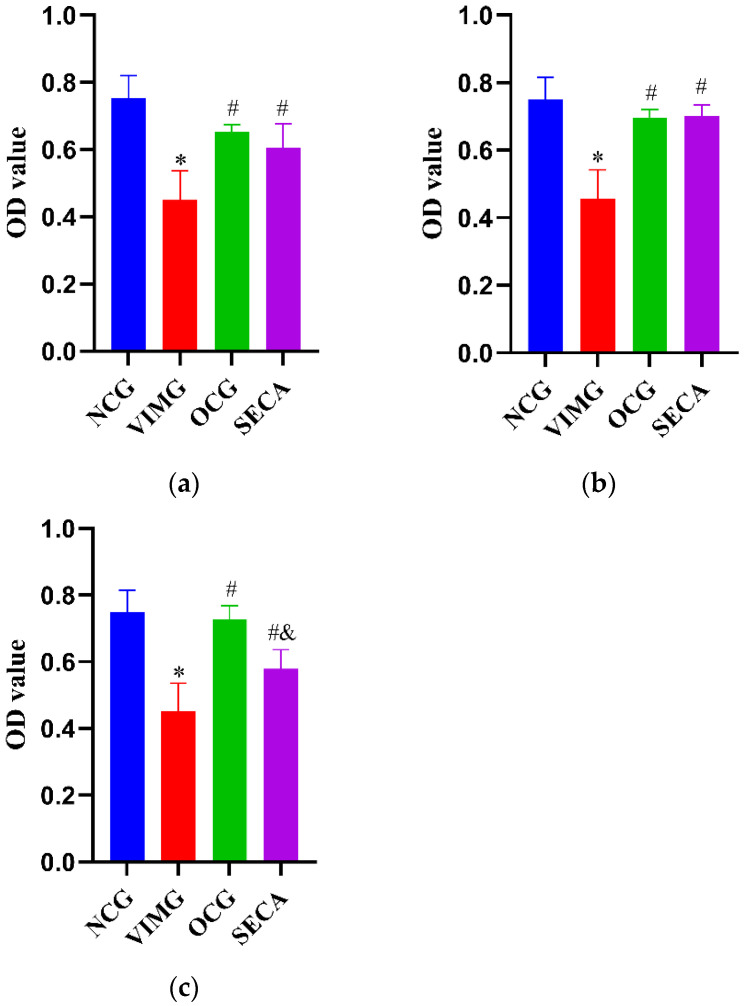
Effect of SECA on H1N1 via premixed administration (**a**), prophylactic administration (**b**), and therapeutic administration (**c**). Comparison with the negative control group (NCG), * *p* < 0.05. Comparison with the virus-infection model group (VIMG), ^#^
*p* < 0.05. Comparison with the Oseltamivir control group (OCG), ^&^
*p* < 0.05. Data were means ± SD of three independent experiments.

**Figure 4 molecules-27-08313-f004:**
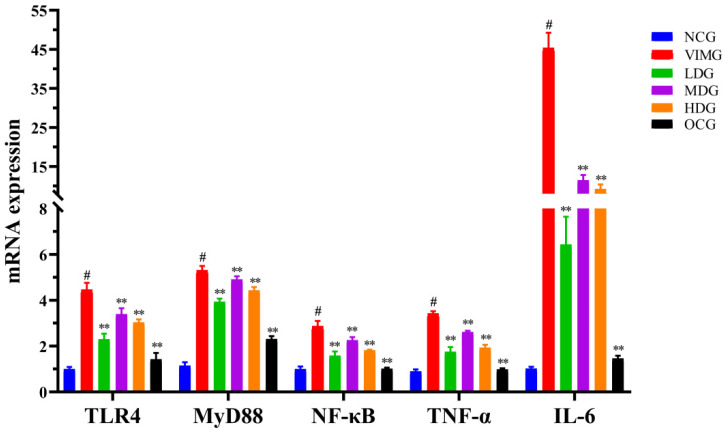
Comparison of relative expressions of TLR4, MyD88, NF-кB, TNF-α, and IL-6 mRNA in each group. Comparison with the virus infection model group (VIMG), ** *p* < 0.01. Comparison with the negative control group (NCG), ^#^
*p* < 0.01. Data were means ± SD of three independent experiments.

**Figure 5 molecules-27-08313-f005:**
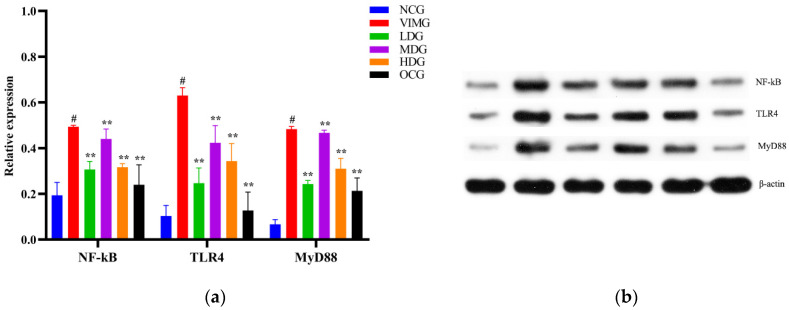
Relative expression of TLR4, MyD88 and NF-кB proteins versus β-actin in MDCK cells (**a**); Detected expression of TLR4, MyD88, NF-кB and β-actin proteins in MDCK cells using Western-blotting assay (**b**). Comparison with virus infection model group (VIMG), ** *p* < 0.01. Comparison with the negative control group (NCG), ^#^
*p* < 0.01. Data were means ± SD of three independent experiments.

**Table 1 molecules-27-08313-t001:** The cytotoxicity of SECA on MDCK cells.

The Concentration of SECA (μg/mL)	Inhibition Rate (%)
50	66%
25	71%
12.5	31%
6.25	21%
3.125	7%
1.5625	4%

**Table 2 molecules-27-08313-t002:** The infectivity of influenza A virus on MDCK cells.

Dilution of Virus	Number of CPE Wells	Number of Non-CPE Wells	Cumulative Number of CPE Wells	Cumulative Number of Non-CPE Wells	Percentage of CPE Wells
10^−1^	8	0	21	0	100%
10^−2^	8	0	13	0	100%
10^−3^	3	5	5	5	50%
10^−4^	2	6	2	11	15.3%
10^−5^	0	8	0	19	0%
10^−6^	0	8	0	27	0%

**Table 3 molecules-27-08313-t003:** Primer sequence and product length for each gene.

Name	Primer Sequence(5′→3′)	Product Length
TNF-α	Forward: CGAACCCCAAGTGACAAGCC	119 bp
	Reverse: TCTGTCAGCTCCACGCCGTTG	
IL-6	Forward: TGACCCAACCACAGACGCCAG	178 bp
	Reverse: AGGAATGCCCATGAACTACAGC	
TLR4	Forward: TGCCAGAATGATGTCTCCTACCC	192 bp
	Reverse: CTCAGGTCCAGTTTCTCGGTT	
MyD88	Forward: CCTGAGCGTTTTGATGCCTT	100 bp
	Reverse: ACTTCAGCCGATAGTTTGTCT	
NF-κB	Forward: GCACAGACACCACCAAGACCCAC	136 bp
	Reverse: CGGCAGTCTTTCCCCACAAGCTC	
Actin	Forward: ACTTTAGTTGCGTTACACCCT	189 bp
	Reverse: TAAATCCTGAGTCAAGCGCCAA	

**Table 4 molecules-27-08313-t004:** Antibodies applied for current research.

Protein	Catalog Number	Species	Dilution	Company
NF-kB	10745-1-AP	Rabbit	1:2000	Proteintech
TLR4	19811-1-AP	Rabbit	1:750	Proteintech
MyD88	23230-1-AP	Rabbit	1:1000	Proteintech
β-actin	60008-1-Ig	Mouse	1:5000	Proteintech

## Data Availability

Data are contained within this article.

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
