# Peer review of "In Vitro Anti-Influenza A Virus H1N1 Effect of Sesquiterpene-Rich Extracts of Carpesium abrotanoides"

_molecules, 2022, doi:10.3390/molecules27238313_

Round 1

Reviewer 1 Report

In the article Anti-influenza A (H1N1) virus effect of sesquiterpene-rich extracts of Carpesium abrotanoides” Li et al. provide data on the in vitro effect of the plant extract on virally infected MDCK cells. As far as I understood, the authors tested only one extract so the title should be corrected accordingly. The article would be certainly interesting, if it was readable. Unfortunately, I cannot recommend it for publication in its current form for the reasons I enlist below.

Major concerns:

1.      In my view, the biggest problem with this article concerns the design of the experiment described in 3.9 of Methods. The authors say they use three different modes of administration of the drug in order to (1) inactivate the virus; (2) prevent infection with the virus and (3) treat the virus after it has been introduced in cells. In the first two cases the drug is either introduced together with the virus or before viral infection and then given again in the SECA group. In the third case it is only given in the SECA group after infection. Is this correct? I honestly do not understand why the authors think that SECA can inactivate the virus or prevent viral infection when the data in Figure 3 (VIMG group) clearly show that cells have a drop in vitality in the VIMG group in all three conditions. Maybe I misunderstood the idea of this experiment, which means it has to be described much more clearly in the methodological part of the paper. Besides, I would not call these “different modes of administration” but simply “different conditions” since, as I mentioned before, I do not see any proof that the virus has been either inactivated or that infection with the virus was perturbed.

2.      The introduction is too short and insufficient. There is no information about the chosen signaling pathway. How did the authors choose exactly these genes/proteins for their mRNA and protein expression part of the work? References are missing in several sites throughout the manuscript (e.g. In our follow-up investigation, the sesquiterpene-rich extracts was extracted from C. abrotanoides and its anti-influenza A mechanism has been studied.) It would be beneficial to also include information about the cell line, which was chosen for these particular experiments.

3.      The same problem is with the Discussion – it is only focused on the signaling pathway. It looks as if the authors wrote one text and split it in two parts – the first half they call Introduction, the second – Discussion. This is really not serious.

Minor concerns:

1.      The Results section is very difficult to understand. In my view paragraphs 2.2 and 2.3 must be supported with data, which is not the case.

2.      I had to scroll down to the Methods part in order to understand what all these abbreviations – VIMG, OCG, NCG, actually mean. Please, clarify them in both the text and also in the figures and their legends.

3.      It is a mystery what are the groups A-F in Figures 4 and 5.

4.      Legends to the figures should be more informative. For example, in Figure 1 the names of the substances can be given and not just the numbers that the authors use. As mentioned before, in Figures 3-5 all details concerning what is shown in the figures must be given, including the used abbreviations. Otherwise, the reader gets completely confused while trying to understand them.

5.      Figure 4 should be redone so that the high bar for IL-6 does not prevent the reader to clearly see the changes in expression for the other genes under study.

6.      Figures are not cited correctly. For example, Figure 3 that contains three different panels is cited only once at the very end of paragraph 2.4 of the Results section.

7.      English must be extensively improved since the manuscript is almost impossible to be read and properly understood. Here is just one example of a sentence that is almost totally incomprehensible: Under the inactivation of virus, the protective effect of SECA on MDCK cells infected with influenza A (H1N1) under the direct inactivation of the virus administration mode showed that compared with the NCG,the activity of the cells in the VIMG decreased and the OD value decreased significantly (P <0.05),indicating that the virus was successfully infected with MDCK cells.

Author Response

Dear Reviewer,

    Many thanks for the comments. We reviesed the manuscript. The details answers were listed in the file "Details response for reviewer 1", one by one according the comments.

     Thank you very much.

      Sincerely

     Dr. Xiaojiang Zhou (Prof.)

Reviewer 2 Report

The manuscript "Anti-influenza A (H1N1) virus effect of sesquiterpene-rich extracts of Carpesium abrotanoides" investigates the action of sesquiterpenes on the H1N1 virus. The article presents an important theme and has been properly described. It presents methods suited to the objectives and congruent with the conclusion. However, the authors should improve the following points:

1. In the introduction: update data on deaths from influenza, owing to the fact that the data cited are from 2009;

2. In figure 3: indicate the meaning of the abbreviations used (NCG, VIMG...);

3. In figure 4: indicate the meaning of the groups in the legend;

4. Indicate the perspectives and limitations of the study.

Author Response

Dear Reviewer,

     Many thanks for the comments. We reviesed the manuscript. The detailed answers were listed in the file "Detailed response for reviewer 2", one by one  acorrding the comments.

     Thank you very much.

    Sincerely

     Dr. Xiaojiang Zhou (Prof.) 

Round 2

Reviewer 1 Report

I believe that the authors did a very good job with the revision of their manuscript. My major concerns have been properly adderessed and it is absolutely clear now how and why they performed the cell experiments. The representation of the results has also been greatly improved. I would still stress on the fact that SECA showed the greatest effect in the case of prophylactic administration. Also, when you write "influenza A virus (H1N1)", the brackets should be removed, because at some point there is a confusion between IAV and H1N1. In my view, there is still some need for improvement of the English language usage but this will probably be solved with the help of the Editorial office. I can now gladly recommend this work for publication in Molecules.

Author Response

Dear review,

    Many thanks. We revised the manuscript according the comments and provide a point-by-point response. Please see the attachment.

    Sincerely

   Dr. Xiaojiang Zhou (Prof.)
